# SARS-CoV-2 Recombination and Coinfection Events Identified in Clinical Samples in Russia

**DOI:** 10.3390/v15081660

**Published:** 2023-07-30

**Authors:** Ekaterina N. Chernyaeva, Andrey A. Ayginin, Alexey V. Kosenkov, Svetlana V. Romanova, Anastasia V. Tsypkina, Andrey R. Luparev, Ivan F. Stetsenko, Natalia I. Gnusareva, Alina D. Matsvay, Yulia A. Savochkina, German A. Shipulin

**Affiliations:** Federal State Budgetary Institution “Centre for Strategic Planning and Management of Biomedical Health Risks” of the Federal Medical Biological Agency, 119121 Moscow, Russiashipulin@cspmz.ru (G.A.S.)

**Keywords:** SARS-CoV-2, COVID-19, whole genome sequencing, recombination, molecular epidemiology

## Abstract

Recombination is one of the mechanisms of SARS-CoV-2 evolution along with the occurrence of point mutations, insertions, and deletions. Recently, recombinant variants of SARS-CoV-2 have been registered in different countries, and some of them have become circulating forms. In this work, we performed screening of SARS-CoV-2 genomic sequences to identify recombination events and co-infections with various strains of the SARS-CoV-2 virus detected in Russia from February 2020 to March 2022. The study included 9336 genomes of the COVID-19 pathogen obtained as a result of high-throughput sequencing on the Illumina platform. For data analysis, we used an algorithm developed by our group that can identify viral recombination variants and cases of co-infections by estimating the frequencies of characteristic substitutions in raw read alignment files and VCF files. The detected cases of recombination were confirmed by alternative sequencing methods, principal component analysis, and phylogenetic analysis. The suggested approach allowed for the identification of recombinant variants of strains BA.1 and BA.2, among which a new recombinant variant was identified, as well as a previously discovered one. The results obtained are the first evidence of the spread of recombinant variants of SARS-CoV-2 in Russia. In addition to cases of recombination we identified cases of coinfection: eight of them contained the genome of the Omicron line as one of the variants, six of them the genome of the Delta line, and two with the genome of the Alpha line.

## 1. Introduction

The SARS-CoV-2 virus was first detected in China in December 2019 and has since spread around the world. During the pandemic, a system for rapid, high-throughput sequencing of the virus genomes was established, which provided a significant amount of data for analysis by the scientific community. In particular, as of 23 June 2022, the GISAID database contained more than 11.5 million virus genomes with accompanying metadata. Analysis of the genomic variability of pathogens that pose a threat to health makes it possible to trace the ways of infection transmission, allows us to study the molecular mechanisms of the manifestation of pathologies, and also contributes to the development of more effective vaccines and specific diagnostic tests.

It is known that the main mechanism of SARS-CoV-2 evolution is the occurrence of point mutations and insertions/deletions; however, recombination mechanisms contribute to a significant extent to the evolution of coronaviruses [1]. Recombination events result in chimeric genotypes of two or more viral variants infecting the same cell. Thus, new unique variants of genomes appear, containing combinations of mutations from various strains. As a result of this process, viruses with altered indicators of both transmissibility and virulence may appear.

The search for recombinant genomes of the SARS-CoV-2 virus has been carried out by various research groups since the beginning of the pandemic. In one of the first papers on this topic, published in mid-2021, Varabyou et al. identified 225 potentially recombinant genomes, some of which, according to the authors, circulated in the population [2]. In September 2021, a paper was presented that examined 279,000 virus isolates collected in the UK between early 2020 and March 2021 [3]. As a result of the analysis, 16 recombinant sequences were identified in the database, 12 of which were clustered into 4 groups, and the remaining 4 showed signs of mosaicism in the genome structure. It should be noted that, in addition to analyzing the directly assembled sequences, the authors confirmed their findings by evaluating raw read alignments for the reference genome and calculating allele frequencies. In April 2022, during the circulation of the Omicron strain, researchers from China presented a paper reporting the presence of recombination between two variants of the Omicron strain (BA.1 and BA.2) and between different VOC/VOI of the SARS-CoV-2 virus [4] (where VOC—variants of concern, VOI—variants of interest according to the classification of the World Health Organization (WHO)). The authors conducted a comparative analysis of mutations in VOC/VOI and concluded that recombination processes were involved in the evolution of the SARS-CoV-2 virus. Potential recombination between Delta and Beta variants that infected one patient in the Orf1ab and Spike genes was described by He et al. [5]. Recombination sites were discovered by studying the allele frequencies of the two strains at different positions, after which the presence of chimeric sequences was confirmed by Sanger sequencing. In March 2022, recombinant forms of the Delta and Omicron strains were found in the Spike protein in the United States [6]. Bolze et al. examined samples obtained during co-circulation in the United States of the Delta and Omicron variants from November 2021 to February 2022 [7]. An analysis of 29,719 genomes revealed 18 cases of co-infection with these strains, one of which was a source of recombination variants, and 2 cases of infection directly with recombinant viruses.

Some of the detected recombinant strains circulating in the population have been assigned letter identifiers. In particular, XD, XE, and XF [8] became such variants with identifiers. The XD variant genome is a combination of the Delta and BA.1 strains, where a portion of the BA.1 variant genome is inserted into the Delta strain genome at positions 21,643–25,581. The XE variant at positions 1-11,537 contains substitutions characteristic of the BA.1 strain, while the rest of the genome originated from the BA.2 strain. The XF variant arose as a result of recombination between the Delta and BA.1 strains. At positions 1-5386, substitutions characteristic of the Delta variant are found, while the rest of the genome contains mutations of the BA.1 strain. The June 2022 publication also describes other recombinant lines (14 variants: XA-XY) with unique identifiers assigned [9].

To sum up, a number of cases of recombination of the SARS-CoV-2 virus have been described to date in the literature. Often, such sequences were found in samples taken from patients co-infected with two strains of SARS-CoV-2. Our manuscript presents the results of the search and analysis of SARS-CoV-2 recombinant variants and co-infections in cases collected in Russia from the beginning of 2020 to February 2022.

## 2. Materials and Methods

### 2.1. Database of SARS-CoV-2 Nucleotide Variants

A database of nucleotide variants (single nucleotide variants and insertions/deletions) used for viruses’ classification and identification of recombinant variants was created based on various SARS-CoV-2 virus strains sequencing data according to the algorithm described below. From the entire variety of SARS-CoV-2 genomes downloaded from the GISAID database (11,692, 140 genomes available on 5 July 2022) using an average of 1000 genomes per SARS-CoV-2 genetic lineage randomly selected, the overall genomes of 1735 lineages were analyzed. If the number of genomes for a genetic line was less than 1000, then all genomes were included in further analysis. For lines with less than 20 genomes, the analysis was not performed. The genome sequences for each of the lines were saved in separate fasta files. The resulting files contained 679,833 genome sequences for 1451 (84% of the original value) SARS-CoV-2 genetic lineage. For the sequences of each of the fasta files, multiple alignments were constructed with the SARS-CoV-2 reference genome (MN908947.3) using the MAFFT program (version v7.490) and the --keeplength and --addfragments options. The resulting multiple alignment files were processed using standard Python language tools and the pandas library. For each of the lines under consideration, positions different from the reference genome were identified, and the proportions of alternative alleles among the sequences belonging to the same line were calculated. The collected data were saved as a table in a TSV file (Appendix A).

### 2.2. Samples Collection

Nasopharyngeal swabs from patients with confirmed COVID-19 infection were collected from 3 February 2020 to 26 March 2022 by Moscow clinical diagnostic laboratories, with subsequent transfer to the PCR laboratory of the Centre for Strategic Planning of FMBA of Russia for further whole genome sequencing. The total number of analyzed samples was 9336. All samples were collected in a transport medium (Disposable Virus Collection Tubes, CDVCT-1, and CDRICH) suitable for storage of the SARS-CoV-2 virus, transferred to the laboratory within 48 h after collection, and stored at −80 ℃ before RNA isolation. Transportation and storage were in accordance with local regulations for the handling of biologically hazardous specimens. For each sample, the following information was stored: a unique identifier, the date the sample was collected, and the geographic region of origin. The samples included in this study were collected in Moscow and the Moscow region.

### 2.3. SARS-CoV-2 Whole Genome Sequencing

The presence of SARS-CoV-2 RNA in the collected biological sample was confirmed using the AmpliTest SARS-CoV-2 test kit (Centre for Strategic Planning of FMBA of Russia, Moscow, Russia), with further identification of SARS-CoV-2 variants using the AmpliTest SARS-CoV-2 VOC PCR assay (Centre for Strategic Planning of FMBA of Russia, Moscow, Russia) according to the instructions. RNA from positive samples containing Omicron strains was isolated using a Ribo-prep purification kit (Centre for Strategic Planning of FMBA of Russia, Moscow, Russia) and reverse transcribed using an Ampliseq cDNA synthesis kit for Illumina (Illumina; San Diego, CA, USA). The resulting cDNA was amplified using the AmpliSeq for Illumina SARS-CoV-2 research panel (Illumina), which contains 247 amplicons in 2 pools targeting the entire SARS-CoV-2 genome. Library preparation was performed using the AmpliSeq Library PLUS kit (Illumina). Library quality was assessed by capillary electrophoresis using an Agilent 2100 Bioanalyzer system (Agilent; Santa Clara, CA, USA). Library concentration was measured with a Qubit 4 fluorometer (Thermo Fisher Scientific; Waltham, MA, USA) using a Qubit dsDNA HS Assay Kit (Thermo Fisher Scientific, Waltham, MA, USA). Sequencing was performed on an Illumina NextSeq 550 system with a NextSeq 500/550 Mid Output Kit v2.5 (300 cycles) (Illumina). Manufacturers’ recommendations were followed in all cases. The median genome coverage of sequencing reads was 99.68%, with a median coverage per nucleotide of 3851.

### 2.4. SARS-CoV-2 Sequencing of Target Regions

Target Sequencing has been performed using Oxford Nanopore approach. cDNA obtained for whole genome sequencing was used to obtain target fragments of the SARS-CoV-2 genome. Amplification of target fragments was performed using LongAmp^®^ Taq DNA Polymerase (New England Biolabs, Ipswich, MA, USA) and oligonucleotides specific to regions of the SARS-CoV-2 virus genome (pair 1: 5′-TGGTGATTCAACTGAATGCAGCA-3′ + 5′ -AGCTAAAGTTACTG-GCCATAACAGC-3′-2921 b.p.; pair 2: 5′-TGACCAAGACATCAGTAGATT-GTACAA-3′ + 5′-CATGGAGTGGCACGTTGAGAAG-3′-3145 b.p.; pair 3: 5′-TGGTGATTCAACTGAATGCAGCA-3′+ 5′-CATGGAGTGGCACGTTGAGA-AG-3′-3108 bp, pair 4: 5′- TGACCAAGACATCAGTAGATTGTACAA-3′ + 5′-AGCTAAAGTTACTGGCCATAACAGC-3′-2958 bp, pair 5: 5′-TGCACA -AGCTTTAAACACGCTT-3′ + 5′-CGCTAGTAGTCGTCGTCGGTTC-3′-1783 bp, pair 6: 5′-ACGCTTGTTAAACAACTTAGCTCCA-3′ + 5′-TTGGACATG-TTCTTCAGGCTCA-3′-1686 bp, pair 7: 5′-TGCACAAGCTTTAAACACG-CTT-3′ + 5′-TTGGACATGTTCTTCAGGCTCA-3′, 1702 bp; pair 8: 5′-AC-GCTTGTTAAACAACTTAGCTCCA-3′ + 5′-CGCTAGTAGTCGTCGTCGGT-TC-3′, 1767 bp). The preparation of reaction mixtures was carried out according to the manufacturer’s instructions, with the introduction of oligonucleotides at a final concentration of 0.4 uM and 5 μL of the sample after reversion. Amplification was carried out on a SimpliAmp Thermal Cycler (Thermo Fisher Scientific, Waltham, MA, USA) with the following temperature program: 94 °C, 30 s; 40 repetitions: 94 °C—30 s., 62 °C—30 s., 68 °C—3 min.; 68 °C—10 min.

The evaluation of the quality of the resulting amplicon was carried out using agarose gel electrophoresis (2%). Amplicon purification was performed using surface-modified magnetic particles from KAPA Pure Beads (KAPA Biosystems, Wilmington, MA, USA). Measurement of the concentration of nucleic acids in the purified amplicon was carried out using a Qubit 4 fluorometer (Thermo Fisher Scientific; Waltham, MA, USA) and a Qubit dsDNA HS Assay Kit (Thermo Fisher Scientific, Waltham, MA, USA). Library preparation for Oxford Nanopore sequencing was performed with the Ligation Sequencing Kit (SQK-LSK109; Oxford Nanopore Technologies, Oxford, UK). Samples were barcoded using PCR Barcoding Expansion 96 (EXP-PBC096; Oxford Nanopore Technologies, Oxford, UK). Sequencing was performed on an Oxford Nanopore GridION system with a R9.4.1 flow cell. Manufacturers’ recommendations were followed in all cases.

The reads were mapped to the reference Wuhan-Hu-1 genome (MN908947.3) using the bwa mem program [10]. BAM files were generated using the samtools program [11]. The BAM files were analyzed using the Tablet program [12].

### 2.5. Consensus Sequences and Genomic Variant Identification

Sequencing data were generated for a total of 9336 samples. These sequences were then processed according to the algorithm described below. Raw reads were filtered (minimum mean quality >30 and length >50), and adaptors were cut using tool cutadapt (version 2.10) [13] to remove adapter and primer sequences. Low-quality ends were trimmed with prinseq-lite (version 0.20.4) [14]. The truncated reads were mapped to the reference Wuhan-Hu-1 genome (MN908947.3) using the bwa mem program [10]. BAM files were generated using the samtools program [11]. Single nucleotide variations (SNVs) and short insertions and deletions (InDels) were detected using the GATK (version 4.1.9.0) [15] program, which generates .vcf files. SNVs, insertions, and deletions were considered consistent if the MQ value was greater than 50 and the DP value was greater than 1000. Regions that were covered by less than 10 reads were masked as N. Consensus genomes were generated by GATK FastaAlternateReferenceMaker using the resulting vcf files. Sequences with more than 4000 nucleotides marked N were removed from further analysis, resulting in a total of 8961 sequences and their .vcf and .bam files.

### 2.6. Data Preparation and Phylogenetic Analysis

For phylogenetic analysis, a dataset of representative genomes containing 2908 complete genomes of various SARS-CoV-2 lineages was downloaded from the GISAID database (2 August 2022). From the downloaded file, a subset of genomes of the BA.1* and BA.2* lineages were formed, containing 1499 sequences. Three sequences of the complete genomes of the detected recombinants were added to the created subset. The complete genome of the bat coronavirus RaTG13 (genbank: MN996532.2) was used as an outgroup for phylogenetic analysis; the entire genome of Wuhan SARS-CoV-2 (hCoV-19/Wuhan/Hu-1/2019, genbank: MN908947) was used as a control genome in the analysis. In all obtained sequences, some parts of the genome were pre-masked with N due to their insufficient coverage in sequenced samples (10,600–10,650, 14,406–14,410, 21,800–21,850, 23,020–23,084, 29,829–29,879). All genomes were aligned with the Wuhan-Hu-1/2019 reference genome using MAFFT v7.490 [10] and FastTree 2.1.11 [16] using the following commands: FastTree -gtr -nt alignment_file > tree_file. The visualization of phylogenetic trees was performed using the FigTree v.1.4.4 program [17].

### 2.7. Principal Component Analysis

Principal component analysis (PCA) was carried out as follows:Using the VCF files, a table was assembled containing information on the substitutions found in each of the samples;Filtering of substitutions in low-coverage regions (<10 reads) of the genome was carried out;We removed all substitutions that were not in the reference database, as well as substitutions whose occurrence count was less than 10;For each sample, a binary replacement vector was created (each element of the vector corresponds to a certain replacement. The value of the element is 1 if the replacement is present in the sample and 0 if it is absent);Using the obtained vectors, the principal component analysis was carried out and implemented in the scikit-learn library of the Python language;Visualization of the result was performed using the seaborn library.

### 2.8. Algorithm for the Detection of Recombinant Genomes and Co-infections

To search for potential recombinant viral genomes and co-infection cases, we performed an analytical method based on the analysis of sequencing reads aligned on a reference genome and used a database of SARS-CoV-2 genome-wide sequence data and genetic lineage classification. A detailed description of the method is described in the Appendix A.

## 3. Results

### 3.1. SARS-CoV-2 Nucleotide Variations Database

A database of nucleotide variants degerming various SARS-CoV-2 phylogenetic lineages was created using representative genomes from the GISAID database. The database contains 679,833 genomes representing 1451 SARS-CoV-2 genetic lineages, downloaded from the GISAID database on 5 July 2022. The database of SARS-CoV-2 nucleotide variations contains information on 4526 unique substitutions with a frequency higher than 50% in 1451 SARS-CoV-2 genomes. The median number of mutations in a lineage was 16, and the average was 22.6. The frequency of substitutions in the SARS-CoV-2 genome, calculated using the created database, is shown below (Figure 1).

The variability of various viral genome regions was studied. According to Figure 1, the major nucleotide substitutions are focused around coordinates 10,000–12,000 (ORF1ab gene), 21,000–24,000 (Spike gene), 26,000–27,000 (genes E and M), and 28,000–29,000 (genes ORF9 and N). Analysis of nucleotide substitution frequencies in various parts of the genome allows for the suggestion that SARS-CoV-2 recombinant variants might be more likely to be detected if recombination happens closer to the 3′-end of the genome, because in this case a larger number of SNVs that distinguish viral strains can be analyzed.

### 3.2. SARS-CoV-2 Strain Genomic Classification Using PCA Method

Principal component analysis (PCA) was used to identify recombinant sequences. To identify recombinant genomes, all sequences annotated by the Pangolin program [18] as belonging to the lines BA.1* (BA.1 and sublines), BA.2* (BA.2 and sublines), B.1.617.2, and AY.* (sublines of the Delta variant) were selected. Each sample was assigned a vector containing information about the substitutions present in it. The compiled mutation vectors for the selected samples had a dimension of 600. The results of genome analysis by PCA for the considered samples are shown below (Figure 2).

The results of the PCA method allowed us to discriminate three clusters of genomes that belong to BA.1 and BA.1.1, a cluster of line BA.2, and a cluster of the Delta. The genomes of some samples are presented as outliers compared to clusters corresponding to the canonical non-recombinant lines BA.1, BA.2, and the Delta cluster (B.1.617.2 and AY). Thus, the genome of the LQ-24283 sample (obtained on 23 February 2022) is an outlier of BA.1 located between clusters BA.1 and BA.2; the genomes of the LQ-24660 and LQ-24661 samples (both received on 19 March 2022) are outliers of the BA.2 cluster located between BA.2 and BA.1. These samples were marked as potentially recombinant for further analysis.

### 3.3. Analysis of Potential Recombinant Genomes

#### 3.3.1. Analysis of Consensus Genomic Sequences

The analysis of consensus genomic sequences of potential recombinant variants of SARS-CoV-2 was performed using the Nextclade program [18]. The sequences downloaded from the GISAID database (BA.1: EPI_ISL_14153781, EPI_ISL_14157328, EPI_ISL_14157805; BA.2: EPI_ISL_14189751, EPI_ISL_14189762, EPI_ISL_14195) were used as reference genomes of the BA.1 and BA.2 lineages. Genome sequences with information about the substitutions found in the loaded sequences and in potential recombinant variants are presented below (Figure 3).

As a result of the analysis, samples LQ-24660 and LQ-24661 were identified as XAD recombinants: the part of the genome up to position 24,503 corresponds to the BA.2 lineage sequence, while the sequence after position 26,060 has substitutions characteristic of BA.1. It is interesting that LQ-24283 was not annotated as recombinant by the Nextclade program, but the structure of its genome also allows us to distinguish two regions belonging to different lineages. The part of the genome from position 1 to position 24,503 is similar to that of the BA.1 lineage, while from position 26,060, the SNV profile is similar to that of the BA.2 lineage.

A comparative analysis of the found recombinant sequences was carried out with the previously described recombinant genomes described in the work of Focosi et al. [9] (Figure 4).

As can be seen from the figure above, the LQ-24660 and LQ-24661 genomes are more similar to the previously discovered recombinant sequences of the XAE lines, which does not coincide with the annotation of the Nextclade program, using which these samples were assigned to the XAD lineage. At the same time, no identical recombinant genomes were found for sample LQ-24283.

#### 3.3.2. Analysis of Short Read Alignment Files

To confirm recombination sites and the absence of a mixture of strains in the samples mentioned above, analysis of aligned reads at positions distinguishing variants BA.1 and BA.2 was performed according to the data obtained from the analysis of the collected database (Figure 5).

The results of the analysis of the proportion of the sequence reads in each of the positions that allows us to distinguish BA.1 and BA.2 lineages demonstrate the absence of a mixture of several genetic variants in the sample and also allow us to determine that the recombination points in the genomes are located between positions 24,503 and 26,060 (ORF3a, M genes). It is interesting to note that sample LQ-24283 was assigned by the Pangolin and Nextclade programs to variant BA.1, as well as LQ-24660 and LQ-24661, which, according to our analysis, are products of recombination between BA.1 and BA.2 genomes. The list of substitutions in the identified recombinant genomes is provided in the Appendix A.

#### 3.3.3. Analysis of Nanopore Sequencing Data

To confirm the integrity of the genomes of the identified recombinant viruses, sequencing of the suggested recombination sites was performed using the nanopore sequencing method. A pair of primers was selected to enrich the extended genome region from position 23,791 to position 26,712 (length 2921 bp), covering positions 24,130, 24,503, 26,060, and 26,530, located on both sides of the putative recombination point and enabling the differentiation of options BA.1 and BA.2 (Appendix A). Also, three alternative primer pairs shifted relative to the main one by several nucleotides were used for cross-validation of the results obtained (Figure 6).

The genotype of the omicron BA.1 variant at positions 24,130, 24,503, 26,060, and 26,530 is ATCG, respectively; the BA.2 genotype is CCTA in the same positions, respectively. The genotype of sample LQ-24283 at the indicated positions is ATTA (Figure 6A), which corresponds to the BA.1/BA.2 recombination; samples LQ-24660 and LQ-24661-CCCG (Figure 6B,C), which correspond to the BA.2/BA.1 recombination. Thus, using the selected primer pairs, an extended 2921-bp genome region was amplified. for putative recombinant samples, which was subsequently sequenced using the Oxford Nanopore system and confirmed the presence of recombinant genotypes BA.1/BA.2 and BA.2/BA.1 in the studied isolates.

#### 3.3.4. Phylogenetic Analysis

Using the detected recombinant samples and a representative subset of the genomes of the BA.1* and BA.2* SARS-CoV-2 virus lines from the GISAID database (1499 genomes, uploaded on 8 February 2022), a phylogenetic analysis was carried out, the purpose of which was to establish the nearest related genomes for the studied recombinant isolates (LQ-24283, LQ-24660, and LQ-24661). To construct a phylogenetic tree, regions of the genome with low-coverage were preliminarily masked. The resulting tree is shown below (Figure 7).

Isolates LQ-24660 and LQ-24661 are part of a separate branch within the BA.2 genome cluster. It is interesting that the genomes of hCoV-19/Indonesia/JK-CW62-0773/2022, hCoV-19/Slovenia/195823/2022, and hCoV-19/Pakistan/NIH-B59-GOV-S9/2022 are located on the same branch, downloaded from GISAID, and annotated in the database as belonging to the BA.2 lineage. At the same time, the analysis of these sequences using the Nextclade program showed that these samples belong to the XZ recombinant line (recombination between BA.2 and BA.1 after position 26,060). This observation additionally confirms the previously made assumptions about the detected recombination in isolates LQ-24660 and LQ-2466.

Isolate LQ-24283 forms a separate branch in the BA.1 clade, located at the point of separation of the subtree of samples belonging to the BA.2 line, which also indicates in favor of the recombinant origin of the discovered genome. Moreover, the non-clustered branches closest to the sample under consideration contain a number of BA.1/BA.2 recombinant sequences: hCoV-19/Panama/SEQ1200-GMI/2022 (XAF), hCoV-19/Venezuela/Car9661/2022 (XAD), and hCoV-19/Algeria/22459/2022 (XY). Annotated BA.1/BA.2 recombinant sequences are also found among neighboring clades (Appendix A).

Phylogenetic analysis allowed not only to identify the position of studied recombinant isolates on the phylogenetic tree but also to identify a group of candidates missannotated recombinant genomes in the GISAID database and prove it using Nextclade classification.

### 3.4. Detection of Co-infections in Sequencing Data

Recombination of SARS-CoV-2 genomes can occur when the host is infected with multiple viral strains. Thus, tracking co-infections is an important task for predicting recombination events and their frequency. In the presented work, we screened 9336 sequenced genomes to detect cases of coinfection among patients in Moscow and the Moscow region.

To search for cases of coinfection, we developed an algorithm that allowed us to compare nucleotide variants with reference genomes and determine whether a sample belongs to a group of possible co-infections based on the analysis of vcf files. As a result of the screening of sequenced samples, 23 cases of coinfection were identified. Subsequent analysis of histograms of allele frequencies, as well as information on the sample preparation process (excluding the possibility of contamination) made it possible to confirm 14 of them (Table 1).

Based on our analysis, over 50% of coinfection cases contain Omicron lineages as one of the viral variants (eight cases), six cases contain the Delta variant, and two cases contain the Alpha variant. The most frequent pair of variants is BA.1/BA.2 of the Omicron group of strains (four cases). Also, it is interesting to note that according to Pangolin annotation, the total number of sequenced genomes belonging to the Delta variant is 3981, Alpha is 222, and Omicron is 927. The representation of viral variants in the database might also influence the accuracy of the detection of coinfection cases.

## 4. Discussion

The study of SARS-CoV-2 genomic variability is directly related to the classification of viral genome variants, phylogenetic analysis, and the determination of the phylogenetic lineage, which includes non-recombinant circulating strains. The classification of viral strains is intended to systematize knowledge about common strains, is based on detectable genomic variability, and allows virologists and epidemiologists to track transmission routes and accurately communicate information about circulating variants in different regions of the world. In the study of the COVID-19 pandemic, at least four naming systems for genetically different types and lines of SARS-CoV-2 have been developed and are being implemented: three classifications used in scientific research (Pango, Nextstrain, and GISAID) [19] and the classification proposed by the WHO to identify common variants related to VOI and VOC [20]. Each of the systems has its own scientific approach, all classifications except the WHO classification were proposed before the definition of VOI and VOC. In our work, we relied on the Pango classification, which is based on the analysis of nucleotide substitutions across the SARS-CoV-2 genome. The phylogenetic lineage algorithm underlying the Pangolin Most Likely Phylogeny Lineage Determination Tool [16] is based on machine learning methods and comparative analysis of the studied genomic sequence with a database of tagged genomes. This method has certain limitations and is able to identify recombinant genome variants if they are already present in the Pangolin training dataset.

To identify new variants of recombinant forms of SARS-CoV-2, as well as to search for cases of simultaneous infection of a patient with different strains, we created a database of genomic variants characterizing 1451 genetic lineages of the virus. Using this database, we carried out our classification. As a part of the study, we analyzed the genomic sequences of strains collected from Russian patients diagnosed with COVID-19, which were compared with a database of genomic variants characterizing SARS-CoV-2 genetic lines annotated using the Pangolin algorithm. Any inaccuracy in the classification of non-recombinant lines may be due to the analytical tools available today.

The analysis made it possible to detect several cases of infection of Russian patients with recombinant variants of SARS-CoV-2: two cases of infection with a recombinant XAD variant according to the Pangolin or XAE classification based on the publication of Focosi et al. [9], which is a case of recombination of BA.2 and BA.1 lines (isolates LQ-24660 and LQ-24661), as well as one case of infection with a new recombinant variant BA.1/BA.2 (isolate LQ-24283) with localization of the recombination point between genome coordinates 24,503 and 26,060 (ORF3a, M genes). All three variants of the recombination were discovered in the spring of 2022, during the period of high prevalence of SARS-CoV-2 in Russia as well as around the world. A detailed phylogenetic analysis of the studied population of SARS-CoV-2 viruses and a representative sample of GISAID revealed new recombinant variants of the genomes identified by GISAID as BA.2 variants (Table.1). It is important to understand that the use of phylogenetic analysis for the determination of potential recombinant forms of SARS-CoV-2 could not be used as the only method due to the small number of recombinant genome variants in the GISAID reference database, which can lead to unambiguous results and some mistakes in classification. It is necessary to use alternative approaches to detect previously known SARS-CoV-2 and search for new recombinant forms.

Recombination of viruses in the coronavirus group is a common phenomenon and one of the mechanisms of evolution [21]. Recombinant variants of SARS-CoV-2 are increasingly being discovered by scientific teams due to the accumulation of genomic data [9]. A possible effect of the SARS-CoV-2 recombination process might be the emergence of virus variants with new properties, which is due to the biological effect of individual nucleotide substitutions and their combinations.

Also, as a result of the analysis, 14 cases of coinfection were found, most of which (eight samples) contained the Omicron strain as one of the variants. The total number of genomes of the Omicron strain sequenced in the laboratory was four times less compared to the Delta variant, which was present in six samples. This result may be due to the higher transmissibility of this variant compared with the lines found earlier and the greater likelihood of infection with two or more strains. Apparently, this fact also explains the higher detection of recombinants of the BA.1/BA.2 species in comparison with the previously studied VOC/VOI.

The spread of recombinant variants of SARS-CoV-2 has become widely discussed recently due to their significant role in the epidemiological process. For example, starting in October 2022, sublines of the XBB* variant (a recombinant variant of Omicron BA.2.10.1 and BA.2.75) will be widely spread across the world [22]. Our data also suggest that the ability of Omicron variants for rapid distribution and mutations to result in resistance to immune defenses acquired upon infection with earlier SARS-CoV-2 variants is likely to contribute to the emergence of more and more frequent cases of coinfection as well as the emergence of new recombinant forms. However, not all recombinant forms play an important role from an epidemiological point of view and could be classified using standard methods for determining the genetic lineage of SARS-CoV-2.

## Figures and Tables

**Figure 1 viruses-15-01660-f001:**
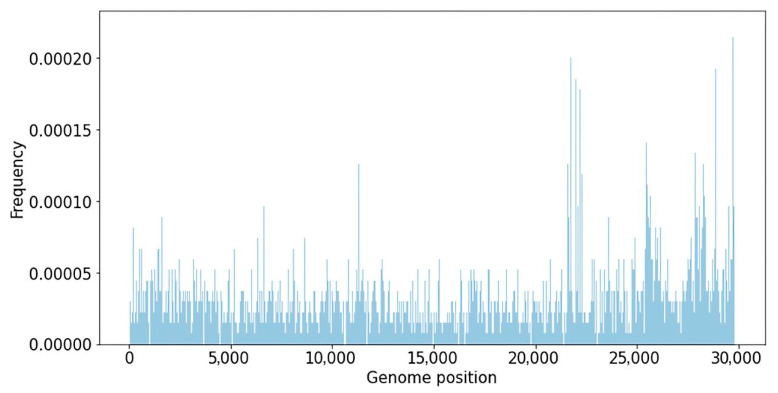
Mutation frequency in various phylogenetic lineages of SARS-CoV-2 across the genome.

**Figure 2 viruses-15-01660-f002:**
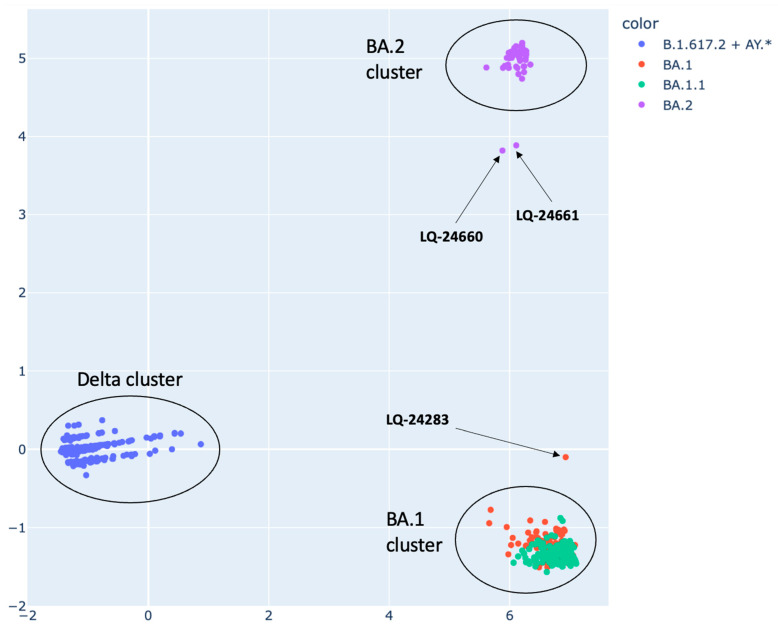
Graph describing principal component analysis (PCA) results for the analysis of SARS-CoV-2 isolates belonging to the phylogenetic lineages BA.1*, BA.2, B.1.617.2 and AY.*.

**Figure 3 viruses-15-01660-f003:**
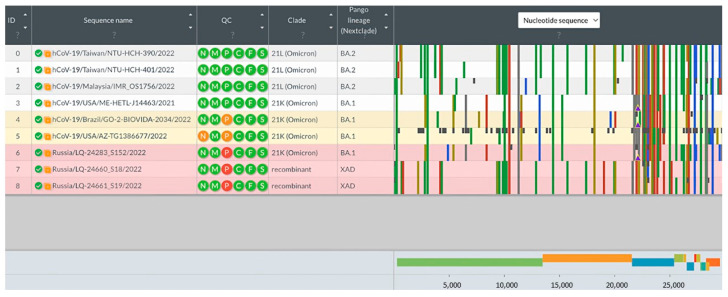
Visualization of the analysis for potential genomic recombinant identification based on SARS-CoV-2 whole genome sequences using the Nextclade program.

**Figure 4 viruses-15-01660-f004:**
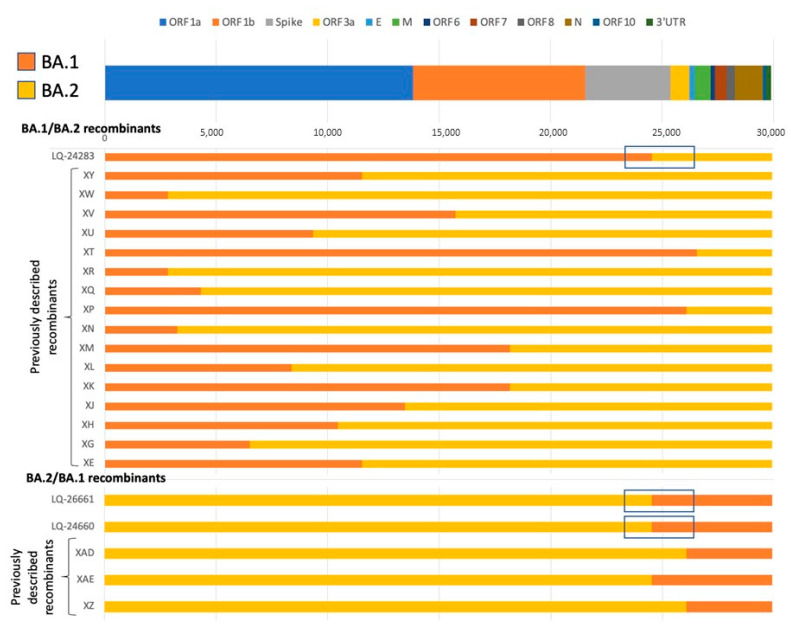
Comparative analysis of previously described recombinant sequences and tested samples from the Moscow region: LQ-24283, LQ-24660, and LQ-26601. Blue frames indicate putative recombination points.

**Figure 5 viruses-15-01660-f005:**
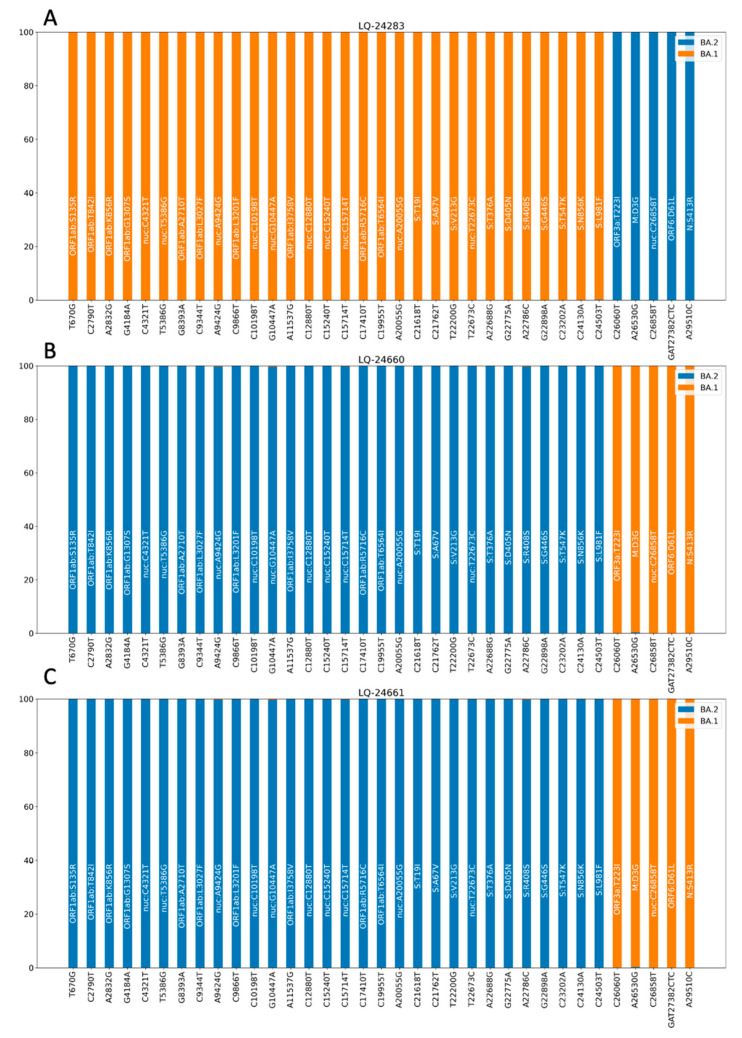
Proportion of reads from WGS data detecting SNPs in the genome of recombinant SARS-CoV-2 variants. The analysis performed on the BAM files shows genomic variants characteristic of the BA.1 and BA.2 lines. Orange highlights present/absent substitutions characteristic of BA.1, blue is characteristic of BA.2. (**A**)—sample LQ-24660; (**B**)—sample LQ-24661; and (**C**)—sample LQ-24283.

**Figure 6 viruses-15-01660-f006:**
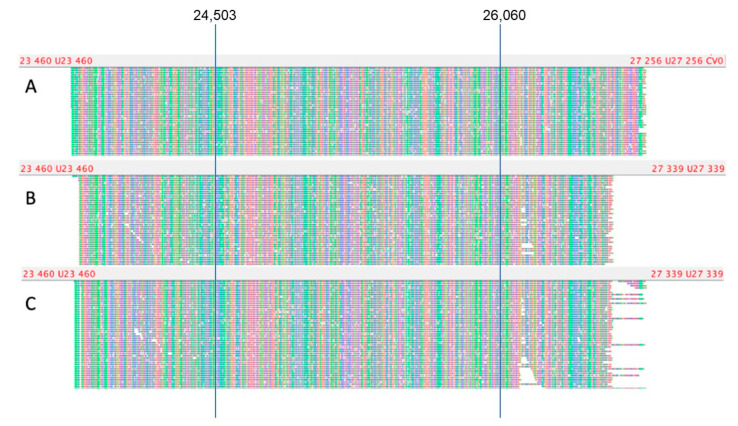
Visualization of sequencing of an extended genome region containing the putative recombination point for samples LQ-24283 (**A**), LQ-24660 (**B**), and LQ-24661 (**C**).

**Figure 7 viruses-15-01660-f007:**
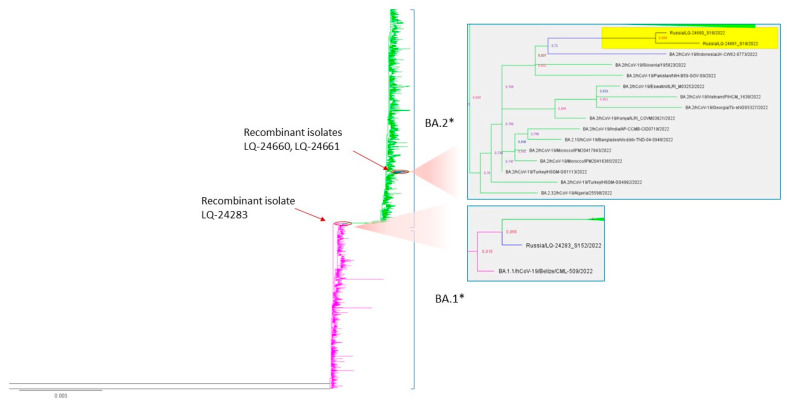
A phylogenetic tree was constructed using the genome sequences of the BA.1* and BA.2* variants from a representative GISAID dataset and three genomes of the recombinant variants identified in our study: LQ-24283, LQ-24660, and LQ 24661. Recombinant samples in the BA.2 clade from Russia are highlighted in yellow.

**Table 1 viruses-15-01660-t001:** Description of detected cases of coinfection in one patient with different strains of SARS-CoV-2.

Sample	Detected Genetic Lines	Date of Sampling
LQ-23247	Delta/21J, Omicron/BA.1	29 January 2022
LQ-21013	Omicron/BA.1, Omicron/BA.2	08 January 2022
LQ-23061	Omicron/BA.1, Omicron/BA.2	26 January 2022
LQ-23066	Omicron/BA.1, Omicron/BA.2	26 January 2022
LQ-21871	Omicron/BA.1, Omicron/BA.2	19 January 2022
LQ-22654	Delta/21J, Omicron/BA.1	23 January 2022
LQ-24255	Omicron/BA.2, Omicron/B.1.1.529	22 February 2022
GCG-1605	Delta/21J, Omicron/BA.1	25 February 2022
LQ-3635	Alpha/B.1.1.7, Beta/B.1.351	02 May 2021
LQ-3188	Alpha/B.1.1.7, Beta/B.1.351	23 April 2021
LQ-3937	B.1.1, Delta/AY.122.4	07 May 2021
LQ-6578	Delta/AY.17, Delta/AY.121.1	01 July 2021
LQ-6061	Alpha/B.1.1.7, Delta/B.1.617.2	28 June 2021
KAB	B.1.232, B.1.1.523	05 April 2021

## Data Availability

SARS-CoV-2 sequences studied in current research have been uploaded to GISAID under the submitter names “Federal State Budgetary Institution” and “Centre for Strategic Planning and Management of Biomedical Health Risks”.

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
