# Peer review of "SARS-CoV-2 Recombination and Coinfection Events Identified in Clinical Samples in Russia"

_viruses, 2023, doi:10.3390/v15081660_

Round 1

Reviewer 1 Report

The manuscript authored by Chernyaeva EN – Ayginin AA et al describes  SARS-CoV-2 recombination and coinfection events in clinical samples from Russia.

The authors sequenced a large number of samples from Moscow and used several robust analysis methods for which they applied precautionary conditions. They identifed three recombinant sample sequences involving the BA.1 and BA.2 strains, that were confirmed by nanopore-sequencing by specific amplicons spanning targeted regions. They also confirmed fourteen cases of coinfection from which half contained the omicron variant.

The author’s findings contribute to the knowledge in the field, but I note a major inconsistency regarding the omicron variants and an excessive technical orientation of the manuscript’s content.

The authors should describe how many strains have been sequenced in total and it would be interesting to see the volution of the SARS-CoV-2 strians in general in the Russian epidemic (or at least in moscou, where the strains come from). Having this information, it is also easier to understand the recombination patterns and patterns of dual infections.  It allows also to see the percentage of dual infections and recombinants in the epidemic to evaluate a potential role of recombinants in future epidemics. 

- Omicron variants:

The authors performed the screening of genomic sequences to identify recombination events and co-infections with various strains of the SARS-CoV-2 virus detected in Russia from February 2020 to March 2022, as indicated in the abstract and at the end of the introduction section. But in the Materials and Method section, paragraph 2.3 they wrote that libraries preparation was undertaken on RNA from positive samplescontaining Omicron strains (nothing about the other variants). This should be clarified because: 1- omicron strains did not emerge before the end of 2021, 2- from the 14 cases of coinfection, 6 samples did not contain any omicron variant. 

- The manuscript contains many technical details in the Materials and Methods and in the Results sections. This is mandatory regarding the scope of the paper, but this results in a reading a bit long. I wonder if it would not be possible to shorten the manuscript and relocate some methods and/or results into the supplementary section. For example, paragraph 2.8 could be given as supplementary method, and include the table S2. A second example could be the Table 1, also.

- The title of the manuscript is imprecise and should contain ‘in Russia’, since this study does not involve the screening of all the sample sequences deposited in databases, but focuses on samples from Moscow.

- Paragraph 2.4: nanopore sequencing of targeted regions of recombination:

The authors did not specify anything about the libraries preparation (which kit was used, if multiplexing was applied) and also which platform was used (MinION, Gridion..).

- Paragraph 3.3.3 analysis of nanopore sequencing data:

In the text the authors speak about Figures 5A, 5B, 5C and the legend of the figure indicates Figure 6. Please correct.

Also, the authors should add some arrows to better show the variants genotypes at several positions, this is not readable in the figures.

the authors should discuss on recombinants and dual infections more in a global context of the epidemic. 

only minor edits

Author Response

We would like to thank for valuable comments. All suggestions has been accepted.

“The authors should describe how many strains have been sequenced in total and it would be interesting to see the evolution of the SARS-CoV-2 strains in general in the Russian epidemic (or at least in moscou, where the strains come from). Having this information, it is also easier to understand the recombination patterns and patterns of dual infections.  It allows also to see the percentage of dual infections and recombinants in the epidemic to evaluate a potential role of recombinants in future epidemics”. 

Agree and the text is changed, We added the number of sequenced and analyzed strains has been added, also we gave some text in discussion section concerning prevalence of recombination events and coinfection among rapidly spread Omicron strains. Description of SARS-CoV-2 evolution in general will be a subject of our next paper that we are preparing now.

“Omicron variants:

The authors performed the screening of genomic sequences to identify recombination events and co-infections with various strains of the SARS-CoV-2 virus detected in Russia from February 2020 to March 2022, as indicated in the abstract and at the end of the introduction section. But in the Materials and Method section, paragraph 2.3 they wrote that libraries preparation was undertaken on RNA from positive samplescontaining Omicron strains (nothing about the other variants). This should be clarified because: 1- omicron strains did not emerge before the end of 2021, 2- from the 14 cases of coinfection, 6 samples did not contain any omicron variant” 

Argee. We analyzed all SARS-Cov-2 variants, not only Omicron. The mistake is changed.

“The manuscript contains many technical details in the Materials and Methods and in the Results sections. This is mandatory regarding the scope of the paper, but this results in a reading a bit long. I wonder if it would not be possible to shorten the manuscript and relocate some methods and/or results into the supplementary section. For example, paragraph 2.8 could be given as supplementary method, and include the table S2. A second example could be the Table 1, also”.

Agree. Changed

“The title of the manuscript is imprecise and should contain ‘in Russia’, since this study does not involve the screening of all the sample sequences deposited in databases, but focuses on samples from Moscow”.

Agree. Changed

“Paragraph 2.4: nanopore sequencing of targeted regions of recombination:

The authors did not specify anything about the libraries preparation (which kit was used, if multiplexing was applied) and also which platform was used (MinION, Gridion..)”.

Agree. Methods are described

“Paragraph 3.3.3 analysis of nanopore sequencing data:

In the text the authors speak about Figures 5A, 5B, 5C and the legend of the figure indicates Figure 6. Please correct”.

Corrected

.

“Also, the authors should add some arrows to better show the variants genotypes at several positions, this is not readable in the figures”.

The figure is corrected, the target region is marked with lines. 

“the authors should discuss on recombinants and dual infections more in a global context of the epidemic”. 

Agree. Discussion section is corrected and more recent data are added.

Reviewer 2 Report

This article “SARS-Cov-2 recombination and coinfection events identified in clinical samples” is about an interesting and recently topic. The article is extensive and methodically explains each of the methods used. It also obtains logical and well-argued results.

However, there are some points to improve, such as:

1.       they must include in section 2.2  the number of samples studied and subsequently how many of them obtained a complete sequence to carry out the study.

2.       in figure 5 the names of the samples in the figure and the one explained in the figure foot are wrong

3.       Figure 7 must be improved.

4.       As a suggestion table 1 can be supplementary material

5.       Some typing errors:

a.       Line 400: LQ-24661

b.      Line 457: “we” twice

Author Response

Thank you for the comments and attention to detail, we have made changes to the text and tried to improve our figures.

  1. they must include in section 2.2  the number of samples studied and subsequently how many of them obtained a complete sequence to carry out the study.

Agree. The text is corrected. We analyzed 9,336 whole genome sequences of the COVID-19 pathogen obtained as a result of high-throughput sequencing on the Illumina platform.

  1. in figure 5 the names of the samples in the figure and the one explained in the figure foot are wrong

agree. changed

  1. Figure 7 must be improved.

We improved the quality

  1. As a suggestion table 1 can be supplementary material

Agree, changed

  1. Some typing errors:

 Line 400: LQ-24661

  Line 457: “we” twice

      Changed